# Dynamic Intercellular Networks in the CNS: Mechanisms of Crosstalk from Homeostasis to Neurodegeneration

**DOI:** 10.3390/ijms26178155

**Published:** 2025-08-22

**Authors:** Yutian Zheng, Rui Huang, Jie Pan

**Affiliations:** 1San Domenico Upper School, San Anselmo, CA 94960, USA; 2Stanford Graduate School of Education, Stanford, CA 94305, USA; 3Department of Pathology, Stanford University School of Medicine, Stanford, CA 94305, USA

**Keywords:** crosstalk, CNS, demyelinating diseases, aging, Alzheimer’s disease

## Abstract

Intercellular communication in the central nervous system (CNS) is essential for maintaining neural function and coordinating responses to injury or disease. With recent advances in single-cell and spatial transcriptomics, a growing body of research has revealed that this communication is highly dynamic, shifting across states of health, aging, demyelination, and neurodegeneration. In this review, we synthesize the current findings on intercellular communication networks involving neurons, astrocytes, microglia, oligodendrocytes, and other glial populations in the CNS across four major states: healthy homeostasis, aging, demyelinating diseases, and Alzheimer’s disease (AD). We focus on how changes in intercellular communication contribute to the maintenance or disruption of CNS integrity and function. Mechanistic insights into these signaling networks have revealed new molecular targets and pathways that may be exploited for therapeutic intervention. By comparing the intercellular signaling mechanisms across different disease contexts, we underscore the importance of CNS crosstalk not only as a hallmark of disease progression, but also as a potential gateway for precision therapy.

## 1. Introduction

The central nervous system (CNS) is composed of a diverse set of cell types—including neurons, astrocytes, oligodendrocytes, microglia, and vascular cells—that engage in tightly regulated intercellular communication to maintain neural homeostasis and coordinate responses to physiological changes. While neurons transmit signals via synaptic connections, glial and vascular cells play equally critical roles by supporting metabolic functions, maintaining immune surveillance, and modulating neuronal activity [1,2,3]. Increasing evidence suggests that these populations form a highly integrated communication network essential for CNS function [4,5,6].

Among the various modes of intercellular communication, ligand–receptor interactions represent a central and highly versatile mechanism by which CNS cells exchange signals in both physiological and pathological contexts. These interactions orchestrate a wide range of cellular processes—including immune responses, trophic support, and tissue remodeling—and are dynamically regulated across states such as aging, demyelination, and neurodegeneration. Dysregulation of ligand–receptor signaling has been implicated in neuroinflammation, synaptic dysfunction, and glial reactivity, making it a critical pathway for understanding disease progression and therapeutic intervention. In this review, we focus specifically on ligand–receptor-mediated communication, aiming to systemically compare how these signaling interactions are altered across four major CNS states: heathy homeostasis, aging, demyelinating conditions, and Alzheimer’s disease (AD).

While many studies have investigated cell-type-specific functions in isolated CNS states—such as neuroinflammation in aging or glial activation in AD—a critical gap remains in our understanding of how ligand–receptor signaling networks are reconfigured across diverse biological contexts in a comparative and integrative manner. Most of the existing reviews focus on either a single disease model or a limited subset of cell types, often without incorporating recent advances in high-resolution transcriptomics. Moreover, the integration of spatial context and dynamic ligand–receptor signaling data across multiple conditions has not been systematically addressed. As a result, our knowledge of conserved versus context-specific signaling pathways, and their roles in CNS resilience or vulnerability, remain fragmented.

Recent advances in single-cell and spatial omics technologies have revolutionized our ability to investigate cellular heterogeneity and intercellular communication in the CNS [7,8,9]. Traditional bulk transcriptomic analyses, while informative, mask the contributions of individual cell types and obscure spatial organization. While traditional bulk transcriptomic analyses have provided valuable insights into global gene expression changes, they inherently mask the contributions of individual cell types, obscure spatial organization, and fail to capture cellular heterogeneity. These limitations have hindered a precise understanding of cell-type-specific mechanisms and microenvironmental interactions, particularly in complex tissues such as the brain. Such gaps underscore the growing need for high-resolution technologies that can resolve gene expression at the level of individual cells and their spatial contexts. In contrast, single-cell RNA sequencing (scRNA-seq) provides gene expression profiles at the level of individual cells. This allows researchers to uncover rare or transient cell states and to reveal the transcriptional diversity across both neuronal and non-neuronal populations. More recently, spatial transcriptomics techniques, such as Nanostring [10], MERFISH [11], TF-seqFISH [12], 10x Visium, and 10x Xenium [13,14], have added a critical spatial dimension to transcriptomic profiling, allowing researchers to map gene expression back to its tissue context without the need for cell dissociation. These high-resolution technologies have made it possible to infer the putative ligand–receptor interactions between cell types, reconstruct signaling networks, and identify changes in communication dynamics across physiological and pathological states. Computational frameworks, such as CellPhoneDB [15], NicheNet [16], and CellChat [17], have further enhanced these efforts by integrating transcriptomic data with curated databases of signaling molecules and pathways to model how cells influence each other in situ. Importantly, spatial omics approaches have provided complementary insights by preserving the anatomical localization of signaling events, which is essential for interpreting region-specific or niche-specific cell–cell crosstalk in the CNS [18].

In this review, we present a comprehensive synthesis of recent findings on CNS intercellular communications across four major states: heathy homeostasis, aging, AD, and demyelination. By leveraging the insights from single-cell and spatial transcriptomic technologies, we highlight emerging principles of cell–cell crosstalk, identify the shared and unique communication modules, and discuss their implications for disease mechanisms and therapeutic strategies.

## 2. The Intercellular Communication Landscape in the CNS: Modes and Mechanisms

The CNS functions as an intricate, multicellular ecosystem in which neurons, glial cells (astrocytes, microglia, and oligodendrocytes), and vascular elements engage in constant dialogue to sustain homeostasis and adapt to physiological and environmental demands. These interactions occur via diverse and often overlapping modalities that operate across both spatial and temporal scales. A clear understanding of these communication mechanisms is essential to dissect how they become disrupted in aging and disease contexts. This section provides a foundational framework of canonical intercellular communication modes in a healthy CNS, setting the stage for a comparative analysis across pathological states in the subsequent sections.

### 2.1. Ligand–Receptor Signaling: The Canonical Signaling Backbone

Soluble ligand–receptor interactions represent a central mode of intercellular signaling in the CNS, orchestrating processes from synaptic modulation to immune regulation. Neurons secrete chemokines such as CX3CL1 (fractalkine) to engage CX3CR1 on microglia, modulating their surveillance and activation states [19]. Similarly, astrocyte-derived TGF-β shapes neurogenesis and synaptic pruning [20,21]. These interactions are spatially and temporally regulated, enabling cell-type-specific communication under diverse conditions.

Disruptions in ligand–receptor signaling are implicated in many CNS disorders, including neuroinflammation and synaptic failure, as will be explored in later sections.

### 2.2. Gap Junctions and Metabolic Coupling

Gap junctions, particularly those formed by connexin proteins (e.g., Cx43 in astrocytes) [22], provide direct cytoplasmic bridges between neighboring cells. These junctions support rapid ionic and metabolic exchanges—astrocytes form extensive syncytial networks that buffer extracellular potassium, modulate calcium waves, and sustain white matter energetics [23].

These channels can become impaired in aging and demyelination, leading to metabolic isolation and signaling failure.

### 2.3. Ionic and Metabolic Exchanges

Beyond physical channels, astrocytes and neurons engage in metabolic shuttles, including the glutamate–glutamine cycle and astrocyte–neuron lactate transfer [24]. This exchange regulates neurotransmission and protects against excitotoxicity [25].

Metabolic uncoupling has been implicated in cognitive decline and synaptic vulnerability during aging and AD.

### 2.4. Extracellular Vesicles (EVs)

EVs—including exosomes and microvesicles—enable long-range, cargo-mediated communication across the CNS. These vesicles deliver proteins, lipids, and RNAs between neurons, glia, and immune cells [26,27]. Microglial EVs may propagate inflammatory cues, while astrocyte-derived EVs have been shown to support neuroprotection [28,29].

EV cargo composition and function are altered in neurodegeneration, potentially serving as biomarkers or therapeutic vehicles.

### 2.5. Contact-Dependent Signaling

Juxtacrine signaling via membrane-bound ligand–receptor pairs provides highly localized control of cellular behavior. For example, neuronal CD200 maintains microglial quiescence via CD200R [30,31], while Notch–Delta interactions guide oligodendrocyte progenitor cell (OPC) differentiation and astrocyte maturation [32].

In aging and multiple sclerosis (MS), these pathways are often downregulated, leading to chronic glial activation.

### 2.6. Neurotransmitter and Gliotransmitter Crosstalk

Neurotransmitters (e.g., glutamate, GABA) and gliotransmitters (e.g., ATP, D-serine) mediate bidirectional neuron–glial signaling. Astrocytes detect synaptic activity and modulate it through gliotransmitter release; microglia and OPCs also express neurotransmitter receptors, integrating activity-dependent cues into immune or myelination responses [33,34].

Disruptions in neurotransmitter-based crosstalk are implicated in synaptic dysfunction and maladaptive plasticity in both aging and disease.

## 3. Technologies for Mapping Intercellular Communication in the CNS

Understanding intercellular communication in the CNS requires tools that can resolve not only cellular identity but also spatial organization and interactive states. Traditional transcriptomic approaches average the signals across diverse cell types, obscuring the specificity of cellular dialogues [35]. In contrast, emerging single-cell and spatial omics technologies enable high-resolution mapping of communication networks—providing unprecedented insights into how cellular interactions shape tissue structure and function. This section outlines the key technological platforms that underpin our ability to reconstruct intercellular networks in a healthy and diseased CNS.

### 3.1. Single-Cell and Single-Nucleus RNA Sequencing: Resolving Cellular Identity and Communication Potential

scRNA-seq and single-nucleus RNA sequencing (snRNA-seq) technologies enable the identification of transcriptionally distinct populations across the CNS [36,37]. Beyond cell typing, these datasets can be mined to infer the putative intercellular communication by examining the co-expression of ligand–receptor pairs across cell types. Computational frameworks, such as CellPhoneDB, NicheNet, and CellChat, have reconstructed signaling networks that have revealed, for example, microglia–astrocyte crosstalk in neurodegeneration [38] and OPCs’ responses to environmental cues [39]. These approaches have uncovered both conserved modules and condition-specific rewiring in CNS signaling networks.

In the subsequent sections, we will build upon these tools to dissect how such communication networks shift across physiological and pathological states.

### 3.2. Spatial Transcriptomics: Embedding Communication into Anatomical Context

While single-cell profiling offers molecular granularity, it lacks spatial information—critical for understanding context-specific cell–cell interactions in structured tissue like the CNS. Spatial transcriptomics platforms, including Stereo-seq, Nanostring, MERFISH, 10x Visium, and Xenium, overcome this limitation by combining gene expression analysis with tissue architecture.

In CNS studies, spatial transcriptomics has revealed region-specific intercellular interactions, such as astrocyte–endothelial signaling in perivascular zones and microglia–neuron contact in cortical layers undergoing remodeling [40,41]. Importantly, pathological ligand–receptor interactions (e.g., TREM2-APOE, CD74-APP) often localize to specific regions in disease models such as AD [42].

These insights emphasize that intercellular communication is not only cell type specific but also spatially organized—an essential consideration when interpreting communication reprogramming in disease.

### 3.3. Integrative Frameworks: Linking Cell Identity, Space, and Function

A major advance has been the integration of single-cell and spatial transcriptomic data with functional and structural validation. Combining these omics platforms with immunohistochemistry, live imaging, and electron microscopy has enabled correction of inferred communication hubs with physical contact zones and dynamic cellular behaviors.

For instance, spatial proximity analyses have confirmed that reactive astrocytes encircle degenerating neurons in ALS and MS [43,44,45,46], validating the predictions from ligand–receptor modeling. Temporal profiling (e.g., via longitudinal sampling or pseudotime inference) has further revealed that communication networks evolve over time—e.g., shifting from trophic to inflammatory interactions in neurodegeneration [47].

This multimodal approach has set the foundation for linking communication dynamics to specific outcomes—such as synaptic loss, glial activation, or remyelination—across life spans and disease courses.

## 4. Cell–Cell Communication in a Healthy CNS: A Hierarchical Network of Homeostatic Signaling

In a healthy CNS, a hierarchically organized network of intercellular communication sustains tissue stability, synaptic function, and cognitive performance. This system spans highly localized synaptic crosstalk to tissue-wide homeostatic loops and cross-compartment coordination during development. These interactions are mediated by a diverse set of signaling modes—including ligand–receptor signaling, phagocytic cues, metabolic exchange, and gliotransmission—operating across multiple spatial and temporal scales.

In this section, we highlight representative examples of cell–cell communication at four organizational levels:Multicellular tissue homeostasis;Synaptic refinement;Developmental lineage coordination;Long-range functional integration.

Together, these layers illustrate the spatial and functional diversity of intercellular signaling in a healthy CNS (Figure 1).

### 4.1. Multicellular Crosstalk Supporting Tissue-Level Homeostasis

In an adult brain, CNS homeostasis is actively maintained through reciprocal signaling among neurons, astrocytes, microglia, oligodendrocytes, and vascular cells. This network underpins synaptic transmission, metabolic regulation, and immune surveillance [56].

For example, neurons and astrocytes form metabolic units in which glutamate released during synaptic activity is taken up by astrocytic transporters (e.g., GLT-1), preventing excitotoxicity and recycling neurotransmitters via the glutamate–glutamine cycle [57]. In parallel, astrocytes respond to ATP and norepinephrine through calcium signaling and gliotransmitter release (e.g., D-serine), modulating synaptic strength and circuit excitability [58].

Microglia, through the CX3CL1-CX3CR1 axis, maintain a surveillance state and contribute to synaptic stability [59].

Endothelial cells (ECs), enveloped by astrocytic endfeet, regulate the blood–brain barrier (BBB) permeability and participate in neurovascular coupling via VEGF and angiopoietins signaling [60].

Oligodendrocytes sustain axonal metabolism through the monocarboxylate transporters (MCTs)-mediated lactate shuttle [61].

These multicellular loops form the foundation of CNS stability, and disruptions to these pathways often mark the early stages of pathology.

### 4.2. Localized Glial Control of Synaptic Refinement

At the level of individual synapses, glial cells play a critical role in sculping neural circuits. In the somatosensory cortex, astrocytes engulf excess excitatory synapses via the MEGF10 and MERTK receptors, recognizing “eat-me” signals on synaptic debris [48]. Mice lacking MEGF10 and MERTK retain excess immature synapses, highlighting the importance of astrocytic phagocytosis during development.

This astrocytic pathway parallels microglial pruning via the complement cascade (e.g., C1q/C3), suggesting distributed glial responsibility in shaping synaptic architecture [62].

Such localized signaling mechanisms ensure efficient circuit maturation and prevent aberrant connectivity.

### 4.3. Cross-Compartment Coordination of Lineage Specification

Beyond local interactions, developmental processes depend on long-range, bidirectional communication across cellular compartments. In embryonic spinal cords, neural progenitor cells (NPCs) secrete angiopoietin-1 (Ang1) under Sonic hedgehog (Shh) control, activating Tie2 on ECs. These ECs in turn release TGFβ1, which acts back on NPCs to promote OPCs’ fate via SMAD3 signaling [49].

This example illustrates how vascular components provide instructive cues during neurodevelopment, integrating morphogen gradients and lineage-specific programs.

### 4.4. System-Level Integration in Long-Term Memory Encoding

In the mature brain, intercellular communication underlies not only just homeostasis, but also complex functions like memory formation. In the basolateral amygdala, single-cell and spatial transcriptomics have revealed that memory-encoding neurons maintain Penk^high^/Tac^low^ signatures enriched for neuropeptide and CREB-associated pathways. Simultaneously, adjacent astrocytes undergo transcriptional reprogramming, indicating interdependent responses across cell types [50].

Disrupting astrocytic signaling impairs long-term memory without affecting short-term recall, suggesting that stable memory traces rely on multicellular transcriptional coordination.

This system-level crosstalk emphasizes that cognition is encoded not solely by neurons, but also by supportive glial networks engaged through intercellular signaling.

### 4.5. Integrative Summary: Multiscale Communication in the Healthy CNS

A healthy CNS is maintained through a multiscale communication architecture—where homeostatic, synaptic, developmental, and cognitive processes are coordinated by highly regulated intercellular signaling. These interactions are largely mediated by ligand–receptor pathways, which we will explore in greater details across aging, demyelination, and neurodegenerative diseases in the following sections. An overview of representative intercellular pathways in homeostatic conditions is provided in Table 1.

## 5. Age-Associated Remodeling of Intercellular Communication Networks

Aging profoundly remodels the CNS at the level of cellular composition, function, and intercellular communication. These changes shift the tissue microenvironment from a supportive, homeostatic state toward one characterized by chronic low-grade inflammation (“inflammaging”), impaired metabolic and synaptic support, and compromised vascular integrity. This systemic rewiring affects the glia, neurons, vascular cells, and immune components, collectively disrupting the intricate communication networks that maintain CNS resilience and plasticity. Understanding these alterations is essential for identifying the early mechanisms predisposing individuals to neurodegenerative disease [63,64,65,66] (Figure 1).

### 5.1. Chronic Inflammation and Glial Crosstalk in Aging

A defining feature of CNS aging is “inflammaging”, driven predominantly by microglia and astrocytes adopting reactive phenotypes. Aged microglia upregulate pro-inflammatory cytokines (e.g., IL-1β, TNF-α) and display dysregulated phagocytosis, contributing to a primed state with exaggerated responses to stress [67,68,69]. Transcriptomic profiles have revealed enhanced interferon and complement signaling in these cells.

Concurrently, astrocytes shift toward neurotoxic “A1”-like states, characterized by upregulation of genes such as Serpina3n and C3, and downregulation of homeostatic markers (e.g., Aqp4, Glt1) [70]. This glia–glia feedback loop amplifies inflammation and disrupts neuronal support, setting the stage for synaptic and metabolic dysfunction.

### 5.2. Neurovascular Unit (NVU) Decline and BBB Dysfunction

Aging compromises the NVU, impairing the tightly coordinated communication among ECs, astrocytes, pericytes, and neurons that regulates cerebral blood flow and maintains the BBB. Aged ECs reduce tight junction proteins (Claudin-5, Occludin) and increase adhesion molecules (ICAM1, VCAM1), promoting BBB leakage and peripheral immune infiltration [71,72].

Astrocytic endfeet coverage decreases, alongside a reduced expression of potassium and water channels (Aqp4 and Kir4.1), disrupting iron homeostasis and neurovascular signaling [73]. These deficits impair nutrients delivery and waste clearance, with spatial transcriptomics pinpointing the vulnerability in deep cortical and hippocampal layers, regions often affected early in cognitive decline [38,74,75].

### 5.3. Synaptic Vulnerability and Disrupted Neuron–Glia Communication

Age-related cognitive decline correlates with gradual synapse loss and impaired plasticity. Beyond intrinsic neuronal alteration, dysfunctional neuron–glia crosstalk plays a key role.

Microglia aberrantly reactivate complement-mediated pruning pathways during aging, leading to inappropriate engulfment of functional synapses marked by C1q and C3 [76,77]. Pharmacological blockade of complement component in aged mice rescues synaptic integrity and memory, highlighting the pathological consequence of misdirected pruning.

Aged astrocytes exhibit diminished glutamate uptake (GLT-1/EAAT2) and gliotransmitter release (e.g., D-serine), disrupting NMDA receptor signaling critical for long-term potentiation [78]. These changes contribute to reduced synaptic efficacy and plasticity, reinforcing synaptic fragility.

### 5.4. OPCs Dysfunction and Myelin Integrity Loss

OPCs retain regenerative potential in the adult CNS, contributing to myelin repair. However, aging impairs OPC proliferation and differentiation, reflected by cell cycle arrest signatures, epigenetic repression, and reduced responsiveness to pro-myelinating cues (PDGF-AA, IGF1) [79,80].

Microglia–OPC communication also shifts with age: young microglia support OPC maturation via IGF1 and IL-33, whereas aged microglia secrete pro-inflammatory cytokines (TNF-α and IL-1β) that inhibit OPC differentiation and promote apoptosis [81]. This dysfunction accelerates myelin thinning and white matter loss.

Single-cell multiomic studies have identified “pre-senescent” OPC states with increased Cdkn2a expression and reduced oligodendrogenic enhanced activity, underscoring the intrinsic plasticity loss, compounded by intercellular support [82].

### 5.5. Integrative Single-Cell Atlas Highlights Glial-Centered Aging Signatures

A large-scale brain-wide single-cell atlas from aged mice emphasized glial cells as central players in aging-associated transcriptional remodeling [51]. These datasets revealed increased inflammatory signaling, reduced neuronal structural integrity, and region-specific disruptions in neuropeptide communication, implicating altered glial–neuronal crosstalk in systemic aging phenotypes.

### 5.6. Microglia-Driven Myelin and Cognitive Decline

Microglial aging intrinsically disrupts the oligodendrocyte signaling pathways (e.g., JAM2-JAM3, TNF-NOTCH1), contributing to myelin deficits and cognitive impairment [52]. Accelerated microglial turnover models have demonstrated that microglial dysfunction alone is sufficient to drive white matter pathology, underscoring the microglia’s central role in age-associated CNS decline.

### 5.7. Microglia Maintain Intercellular Balance to Prevent Degeneration

A lifelong microglia deficiency, as in *Csf1r*^ΔFIRE/ΔFIRE^ mice, results in OPC expression, astrocytic osteopontin (Spp1) upregulation, thalamic calcification, and pericyte loss [83]. These phenotypes reverse upon microglia reconstitution, highlighting microglia’s role in sustaining intercellular homeostasis and preventing age-related degeneration.

### 5.8. Immune–Neural Crosstalk Limits Regeneration in Aging Peripheral Nervous System

Beyond the CNS, aging promotes maladaptive neuron–immune interactions. In aged dorsal root ganglia, neurons upregulate CXCL13 and MHC-I, recruiting cytotoxic CD8^+^ T cells that inhibit axon regeneration via caspase-3 activation [53]. Therapeutic deletion of CD8^+^ T cells or CXCL13 neutralization restores the regenerative capacity, illustrating pathological immune–neural signaling with age.

### 5.9. Genetic Studies Highlight Intercellular Communication Pathways in Brain Aging

Genome-wide association studies have implicated immune signaling, synaptic structure, and lipid metabolism pathways—many involved in intercellular communication—as critical determinants of brain aging and neurodegenerative disease risk, emphasizing the centrality of cellular crosstalk disruption in aging phenotypes [84].

### 5.10. Integrative Summary: Aging-Associated Remodeling of CNS Networks

Together, these findings depict aging as a systemic rewiring of CNS intercellular communication, involving glial senescence, neurovascular decline, synaptic dysfunction, and maladaptive immune interactions. The progressive breakdown of these networks diminishes CNS resilience and primes the brain for neurodegeneration, underscoring intercellular communication as a promising target for interventions aimed at preserving cognitive function in aging. Key aging-associated pathways in intercellular signaling are outlined in Table 1.

## 6. Dysregulated Intercellular Communication in CNS Diseases

Precise intercellular communication is essential for CNS homeostasis, enabling synaptic function, metabolic support, and immune surveillance. In neurodegenerative and demyelinating diseases, such as AD and MS, these finely turned communication networks become progressively disrupted. Dysregulated ligand–receptor signaling, lipid metabolism, and inflammatory cascades drive maladaptive processes, including synaptic loss, demyelination, protein aggregation, and chronic neuroinflammation. Recent advances in single-cell and spatial transcriptomics have revealed how specific cellular crosstalk circuits are reprogrammed in disease states, accelerating degeneration and impairing repair.

### 6.1. Alzheimer’s Disease: Glial–Neuronal Crosstalk in Neurodegeneration (Figure 1)

In AD, cognitive decline results from disturbed intercellular signaling that destabilizes neuronal support and promotes pathology. The microglia and astrocytes undergo disease-associated reprogramming: the microglia adopt a disease-associated microglia phenotype largely driven by TREM2 signaling, while the astrocytes shift toward neurotoxic A1 states [70,85]. These glial changes contribute to synaptic stripping, impaired lipid handling, and exacerbate amyloid-beta and tau pathologies. Concurrently, vascular dysfunction compromises the BBB, allowing for peripheral immune infiltration [86].

#### 6.1.1. Microglia–Astrocyte Interactions and Amyloid Plaque Microenvironment

Spatial transcriptomics has revealed that activated microglia accumulate near amyloid plaques, promoting reactive astrocytes phenotypes that disrupt hippocampal neuronal excitability [42]. A ligand–receptor analysis has highlighted perturbed neurotransmitter signaling withing the plaque microenvironment, linking glial crosstalk to synaptic imbalance.

#### 6.1.2. Astrocyte–Neuron Lipid Trafficking Dysfunction

Neuronal lipid clearance via astrocyte transfer is compromised by APOE4, causing lipid droplet accumulation, metabolic stress, and synaptic vulnerability [87].

#### 6.1.3. Microglia–Neuron Lipid Crosstalk and APOE4 Toxicity

APOE4 microglia accumulate lipid droplets and secrete neurotoxic factors, including tau phosphorylation and apoptosis, linking metabolic dysfunction and neurodegeneration [88].

#### 6.1.4. Microglia–Astrocyte Axis in APOE Aggregation and Aβ Plaque Formation

Kaji et al. demonstrated that astrocyte-derived lipidated APOE is internalized by microglia, where it undergoes delipidation in the lysosomes, generating aggregation-prone forms that promote Aβ plaque formation. This finding reveals a key pathway by which astrocyte-to-microglia lipid transport initiates and exacerbates plaque deposition, highlighting intercellular APOE processing as a central event in AD pathogenesis [54].

#### 6.1.5. APOE4 Alters Microglia–Neuron Communication Through Calcium Signaling

Victor et al. found that the microglia in APOE4 mice secrete soluble factors that dampen calcium transients in APOE3 neurons in vitro, disrupting neuronal activity independent of direct contact. This suggests that APOE4-associated microglial reprogramming perturbs synaptic excitability via paracrine signaling, offering a potential mechanism for early synaptic dysfunction in AD [89].

#### 6.1.6. Perivascular Cell–Microglial Communication

Perivascular macrophage- and fibroblast-derived osteopontin (SPP1) license microglia for synaptic pruning in AD, with *Spp1* deletion reducing synapses loss despite an unchanged plaque burden [55].

### 6.2. Demyelinating Disease: Rewiring Communication in Myelin Injury and Repair

Demyelinating diseases, including MS and toxin-induced models (e.g., cuprizone), exhibit profound rewiring of intercellular communication within the neurovascular and glial networks. The early glial responses involve myelin debris clearance and trophic support, but persistent inflammation and immune infiltration often impair remyelination.

Below, we summarize the representative studies that have dissected these intercellular circuits and their implications for remyelination or sustained damage (Figure 2).

#### 6.2.1. Microglia–Oligodendrocyte Lineage Crosstalk

Single-nucleus RNA-seq has identified demyelination-associated oligodendrocytes (DOLs) whose emergence depends on microglial TREM2 signaling [90]. TREM2 and IL-33 coordinate lipid metabolism, phagocytosis, and OPC differentiation, facilitating lesion resolution.

#### 6.2.2. Astrocyte-Derived Cholesterol Supports Repair

Astrocytes at lesion borders upregulate cholesterol biosynthesis via Nrf2, essential for OPC maturation and remyelination; genetic inhibition impairs repair while cholesterol supplementation rescues it [23].

#### 6.2.3. Astrocyte–Microglia Immune Modulation

PD-L1^+^ astrocytes regulate PD-1^+^ microglia to limit inflammation during autoimmune demyelination, illustrating feedback control of glial inflammatory tone [93].

#### 6.2.4. Glial–Immune Cytokine Signaling

IL-3, secreted by reactive astrocytes and infiltrating T cells, activates microglia and myeloid cells, sustaining chronic inflammation in MS [91].

#### 6.2.5. Adaptive Immune Cell Crosstalk in MS

Memory B cells present CNS antigens, such as RASGRP2, to CD4^+^ T cells, perpetuating inflammatory cycles and CNS infiltration [92].

#### 6.2.6. Communication Failure in Leukodystrophy

CSF1R-related leukodystrophy (ALSP) features arrested OPC maturation, dysfunctional microglia, and macrophage infiltration, illustrating how failed glial support drives white matter degeneration [9].

### 6.3. Summary

Dysregulated intercellular communication in AD and demyelinating disease involves maladaptive glial reprogramming, disrupted lipid and immune signaling, and compromised vascular interactions. Single-cell and spatial omics technologies have elucidated the cellular circuits driving pathology and repair, revealing multiple targets for therapeutic interventions. Restoring beneficial crosstalk or interrupting harmful signaling holds promise for mitigating neurodegeneration and promoting CNS resilience. These disrupted pathways are summarized in Table 1.

## 7. Therapeutic Implications of Dysregulated Intercellular Communication in the CNS

Accumulating evidence across diverse CNS disorders supports the notion that dysregulated intercellular communication is not merely a downstream consequence of disease, but a primary driver of pathogenesis. Rather than originating from an intrinsic dysfunction of individual cell types, neurodegenerative and remyelinating conditions often arise from the collapse of coordinated signaling among neurons, glia, vascular cells, and infiltrating immune populations. This systems-level breakdown leads to maladaptive cellular states and microenvironmental imbalances that reinforce disease progression.

Several unifying principles have emerged from recent studies of intercellular interactions in AD, MS, and related disorders:Progressive imbalance between homeostatic and pathological signaling

In early disease stages, compensatory mechanisms—such as lipid recycling, synaptic pruning, and anti-inflammatory signaling—are activated. However, these protective pathways are eventually overwhelmed by chronic inflammation, oxidative stress, and metabolic disruption, reflecting a collapse of intercellular feedback loops that typically maintain tissue homeostasis.

2.Context-dependent disease phenotypes shaped by local microenvironments

Glial cells, such as microglia and astrocytes, adopt diverse transcriptional states depending on the disease stage and spatial contexts, including proximity to lesions, plaques, or the vasculature. These reactive phenotypes, often maladaptive, are driven by aberrant cues from neighboring cells and contribute to ongoing neurotoxicity and repair failure.

3.Disruption of spatial organization in intercellular signaling

Spatial transcriptomics and imaging analyses have shown that disease alters the spatial landscape of cell–cell interactions. Glial clustering, perivascular niche disorganization, and lesion compartmentalization interfere with the signaling networks essential for repair. In demyelinating lesions, the impaired crosstalk among astrocytes, microglia, and OPCs directly contributes to remyelination failure.

These findings underscore the need to target intercellular signaling networks as a therapeutic axis. Rather than focusing solely on the cell-intrinsic pathology or individual molecular targets, future interventions must aim to restore the integrity of multicellular communication that sustains CNS function. Several strategies are being explored:Enhancing protective glia–neuron signaling, including promotion of lipid metabolism and mitigation of excitotoxicity;Reprogramming glial states to favor anti-inflammatory and pro-regenerative phenotypes;Stabilizing neurovascular interactions, such as reinforcing blood–brain barrier integrity and perivascular signaling;Spatially targeted delivery of therapeutic agents to lesion cores and vulnerable tissue niches.

Together, these strategies reflect a paradigm shift in CNS therapeutics—from isolated molecular interventions toward network-level restoration of intercellular coordination. As single-cell and spatial technologies continue to elucidate novel signaling axes, therapeutic success will increasingly depend on our ability to map, interpret, and modulate these dynamic communication networks with spatial and temporal precision.

## 8. Future Directions and Challenges

As single-cell and spatial transcriptomic technologies continue to advance, our understanding of CNS intercellular communication is entering a new era of resolution and complexity. Yet translating these insights into effective, targeted therapies remains a formidable challenge. Future efforts must bridge the gap between observational cell atlases and interventional strategies that manipulate defined signaling axes within specific spatial, temporal, and pathological contexts.

A key direction will be the development of spatiotemporally precise interventions. Most intercellular signals exhibit context-dependent effects—protective in early disease but deleterious in later stages. Thus, static approaches that target entire cell types or pathways are likely to fall short. Instead, emerging strategies aim to manipulate specific cell–cell communication hubs, such as astrocyte–microglia–oligodendrocyte triads, using AAVs, nanoparticles, or ligand–mimetic biologics tailored to region- or state-specific profiles.

Another major frontier lies in multimodal therapeutic platforms, where combinatorial treatments simultaneously modulate multiple communication pathways. For example, integrating lipid metabolism modulators with immunoregulatory agents could reshape the cellular environment more effectively than single-target approaches. However, the complexity of such systems calls for computational frameworks that can predict and optimize their outcomes.

Furthermore, AI-driven communication decoding tools—such as deep learning models trained on spatial ligand–receptor maps—are being developed to identify novel signaling axes and forecast disease trajectories. These tools may enable the development of personalized medicine by revealing patient-specific communication vulnerabilities or drug-responsive networks.

Finally, translating intercellular signaling interventions into a clinical setting will require rigorous validation in human-relevant models. Organoids, xenografts, and precision-cut CNS slices, integrated with multiomic readouts, will be essential to bridge species gaps and evaluate the long-term safety and efficacy.

In summary, decoding and reprogramming the multicellular communication landscape of the CNS offers a transformative path for treating neurodegenerative and demyelinating diseases. Meeting this potential will demand not only technological innovation, but also conceptual shifts—from targeting cell-autonomous pathology to restoring system-level coordination across cell types and compartments.

## 9. Conclusions

Understanding the dynamics of intercellular communication in the CNS has emerged as a key frontier in neuroscience. Far from being static or linear, cell–cell signaling networks in the CNS are highly plastic and responsive to both intrinsic and extrinsic perturbations. Through recent advances in single-cell and spatial transcriptomics, we now appreciate that these interactions are tightly orchestrated in homeostasis, yet undergo profound remodeling in the context of aging, demyelinating disorders, and neurodegeneration, such as AD.

A central insight emerging from recent work is the hub-like role of glial cells—especially microglia and astrocytes—in modulating broader CNS communication. These cells not only respond to cues from injured neurons and the vasculature, but also serve as amplifiers, transducers, or gatekeepers of signaling cascades. Importantly, glial signaling is not uniform; rather, it is regionally compartmentalized, temporally dynamic, and sensitive to both systemic (e.g., immune or metabolic) and local (e.g., synaptic or demyelinating) signals.

Despite the wealth of new data, multiple challenges limit our current understanding. First, most intercellular communication inferences are based on transcript-level predictions, often assuming that ligand and receptor expression correlates with functional signaling—an assumption that ignores post-transcriptional regulation, protein modification, and localization. Furthermore, although tools, such as CellPhoneDB, NicheNet, and CellChat, provide powerful predictions, they remain computationally driven and lack direct functional validation in most settings.

Second, while spatial transcriptomics technologies, like MERFISH, seqFISH+, and Xenium, offer unprecedented anatomical resolution, many lack the throughput or protein-level validation needed for comprehensive signaling reconstruction. Furthermore, the integration of time as a variable—critical for understanding signal initiation, propagation, and resolution—is often absent in current static snapshots. There is a pressing need for longitudinal, multimodal atlases that track intercellular signaling dynamics over time in vivo and in patient samples.

Third, few studies have directly tested whether modulating cell–cell communication pathways can be leveraged therapeutically in a cell-type- and circuit-specific manner. This is a non-trivial challenge: signaling pathways are often pleiotropic, and broadly targeting cytokines or chemokines may produce off-target effects. Nonetheless, some promising strategies are emerging. For instance, targeting the C3-C3aR axis between astrocytes and microglia has shown neuroprotective effects in AD models, while enhancing microglia–OPC communication has been proposed as a strategy to promote remyelination. Similarly, engineered extracellular vesicles (EVs) or receptor decoys may serve as next-generation tools to modulate intercellular signaling with greater specificity.

Moving forward, we envision three key directions for the field:Multiscale Mapping of CNS Communication: Future studies should integrate spatial transcriptomics, proteomics, metabolomics, and functional imaging to build high-resolution atlases that capture CNS signaling across different scales—from molecular gradients to whole-brain networks.Functional Interrogation of Crosstalk Pathways: Genetic and optogenetic perturbation platforms, including CRISPRa/i, DREADDs, and intersectional viral tools, can be leveraged to selectively activate or inhibit specific communication axes in defined cell populations and time windows.Translational Application and Biomarker Discovery: By identifying conserved or disease-specific communication motifs, especially those shared across species or patient cohorts, we can prioritize signaling hubs as therapeutic targets or diagnostic biomarkers. The emergence of spatially resolved, patient-derived data will be crucial in this endeavor.

Ultimately, intercellular communication should no longer be viewed as a static feature of CNS organization, but rather as a dynamic language that cells use to maintain function, respond to stress, and orchestrate repair—or, conversely, to drive dysfunction when misregulated. Deciphering and ultimately manipulating this language holds tremendous promise—not only for understanding brain biology, but for developing precision therapeutics that respect the nuanced cellular choreography of the brain.

**Table 1 ijms-26-08155-t001:** Cell interaction pathways across physiological and pathological states.

Conditions	Pathways	Citation
Health	CX3CL1-CX3CR1 pathway	Zhao et al. [59]
	MEGF10/MERTK phagocytic pathways	Chung et al. [48]
	Ang1-Tie2 pathway	Paredes et al. [49]
	BDNF/MAPK/CREB/ubiquitination/Penk/Tac pathways	Sun et al. [50]
Aging	VEGF/Angiopoietin-Tie/PDGF pathways	Linnerbauer et al. [38]
		Zhang et al. [74]
	JAM2-JAM3/TNF-NOTCH1 pathways	Li et al. [52]
	CXCL13/CXCR5/NFkB/pAKT/pS6 pathways	Rhinn et al. [84]
AD	GABAergic pathway	Mallach et al. [42]
	JAK/STAT pathways	Kaji et al. [54]
	SPP1 phagocytic pathway	De Schepper et al. [55]
Demyelination	TREM2/IL-33/ST2 pathways	Hou et al. [90]
	PD-L1/PD-1 pathway	Linnerbauer et al. [93]
	IL-3/IL-3RA pathway	Kiss et al. [91]
	HLA-DR15/TCR pathways	Jelcic et al. [92]
	CSF1R/CSF1/IL-34 pathway	Pan et al. [9]
	Nrf2/Cholesterol biosynthesis pathways	Molina-Gonzalez et al. [23]

## Figures and Tables

**Figure 1 ijms-26-08155-f001:**
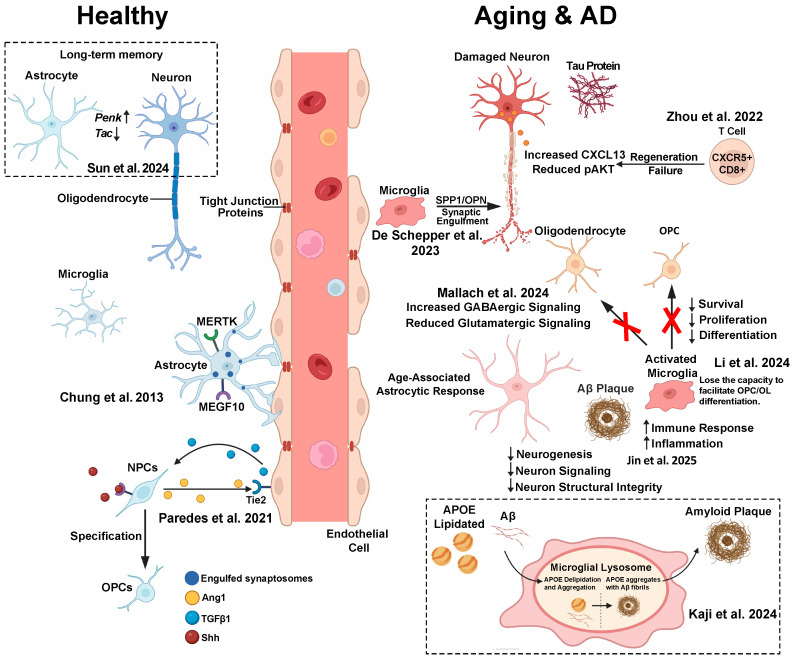
Cell crosstalk model in heathy vs. aging and AD conditions [42,48,49,50,51,52,53,54,55].

**Figure 2 ijms-26-08155-f002:**
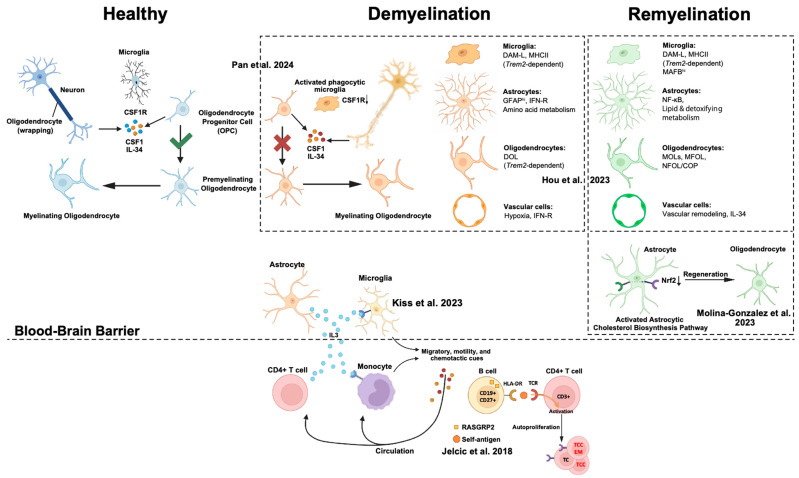
Cell crosstalk model in healthy, demyelination, and remyelination states [9,23,90,91,92].

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
