# Peer review of "Dynamic Intercellular Networks in the CNS: Mechanisms of Crosstalk from Homeostasis to Neurodegeneration"

_ijms, 2025, doi:10.3390/ijms26178155_

Round 1

Reviewer 1 Report

Comments and Suggestions for Authors

Reviewer’s comments

  1. The Introduction – Background & Relevant References in
  • Strengths: The introduction provides a comprehensive overview of CNS intercellular communication, citing key studies (e.g., ligand-receptor interactions, glial roles) and recent advances in single-cell/spatial transcriptomics. References are up-to-date and well-integrated.
  • Suggestions: A brief mention of gaps in prior bulk transcriptomic studies could further justify the focus on high-resolution technologies.
  1. Appropriate Research Design
  • Strengths: The review leverages a robust methodological framework, combining literature synthesis with insights from cutting-edge technologies (scRNA-seq, MERFISH, CellChat). Disease contexts (aging, AD, MS) are logically structured.
  • Note: As a review, the "design" is inherently retrospective, but the selection of studies and technologies is well-reasoned.
  1. Clarity of Presented Results
  • Strengths: Findings are clearly organized by communication modes (e.g., EVs, gap junctions) and disease states. Subheadings enhance readability.
  • Suggestions: Some sections (e.g., aging changes) could benefit from summary tables to compare pathways across conditions.
  1. Conclusions Supported by Results
  • Strengths: Conclusions align well with cited evidence, particularly the therapeutic implications of dysregulated glial-neuronal crosstalk. Limitations (e.g., inferential nature of ligand-receptor predictions) are acknowledged.
  1. Figures & Tables Clarity
  • Strengths: Figures (e.g., models of healthy vs. AD/demyelination) are referenced and enhance understanding.
  • Suggestions: A table summarizing key pathways/dysfunctions could consolidate data.
  1. Originality/Novelty
  • Strengths: The review synthesizes emerging spatial transcriptomics data with classic literature, highlighting novel concepts (e.g., APOE4-driven lipid toxicity, SPP1-mediated synapse loss).
  • Highlight: The focus on "communication as a therapeutic target" is forward-thinking.
  1. Significance of Content
  • Strengths: The article addresses a timely topic with broad implications for neurodegeneration and repair. The emphasis on glial hubs and spatial signaling is particularly impactful.
  1. Quality of Presentation
  • Strengths: Well-structured, with logical flow and professional tone.
  • Minor Issues: A few long sentences could be split for clarity (e.g., Page 1: “ In contrast, single-cell RNA sequencing (scRNA-seq) enables the resolution of gene expression profiles at the level of ndividual cells, uncovering rare or transient cell states and revealing the transcriptional diversity of both neuronal and non-neuronal populations. More recently, spatial transcriptomics techniques, such as Nanostring11, MERFISH12, TF-seqFISH13, 10x Visium and 10x Xenium14, 15, have added a critical spatial dimension to transcriptomic profiling, allowing researchers to map gene expression back to its tissue context without the need for cell dissociation.”).
  1. Scientific Soundness
  • Strengths: Claims are well-supported by primary studies and computational tools. Balanced discussion of limitations (e.g., lack of protein-level validation).
  1. Interest to Readers
  • Strengths: Highly relevant to neuroscientists, neurologists, and therapeutic developers. The translational focus broadens appeal.
  1. Overall Merit
  • Assessment: Excellent scholarly contribution that advances understanding of CNS communication networks. Meets high publication standards.
  1. Quality of English Language
  • Strengths: Clear, concise, and grammatically sound.
  • Minor Edits:
    • Page 1: "transcriptomic" → "transcriptomic analyses" for parallelism.
    • Page 6: "Inflammaging" should be defined at first use.

Summary of Recommendations

  1. Add a table summarizing key communication pathways in health/disease.
  2. Define niche terms (e.g., "inflammaging") early.
  3. Consider splitting long paragraphs for readability.

Recommendation: Minor revision

This is a high-quality, novel review with significant scholarly and translational value. With minor refinements, it would be suitable for a publication in the Int. J. Mol.Sci.

Author Response

Reviewer1:

Reviewer’s comments

1. The Introduction – Background & Relevant References in

  • Strengths: The introduction provides a comprehensive overview of CNS intercellular communication, citing key studies (e.g., ligand-receptor interactions, glial roles) and recent advances in single-cell/spatial transcriptomics. References are up-to-date and well-integrated.
  • Suggestions: A brief mention of gaps in prior bulk transcriptomic studies could further justify the focus on high-resolution technologies.

Response: We thank the reviewer for this helpful suggestion. In the revised manuscript, we have added a paragraph in the introduction that outlines the key limitations of bulk transcriptomic analyses, including their inability to resolve cell-type-specific contributions, spatial organizations, and cellular heterogeneity. We also emphasize how these gaps highlight the need for a high-resolution approaches such as single-cell and spatial transcriptomics. Please find the edited text on Lines 61-73.

2. Appropriate Research Design

  • Strengths: The review leverages a robust methodological framework, combining literature synthesis with insights from cutting-edge technologies (scRNA-seq, MERFISH, CellChat). Disease contexts (aging, AD, MS) are logically structured.
  • Note: As a review, the "design" is inherently retrospective, but the selection of studies and technologies is well-reasoned.

Response: We sincerely thank the reviewer for your positive evaluation of our methodological frameworks and the logical structuring of disease contexts. We are encouraged that the selection of studies and technologies was found to be well-reasoned.

3. Clarity of Presented Results

  • Strengths: Findings are clearly organized by communication modes (e.g., EVs, gap junctions) and disease states. Subheadings enhance readability.
  • Suggestions: Some sections (e.g., aging changes) could benefit from summary tables to compare pathways across conditions.

Response: We thank the reviewer for highlighting the clarity of our organization and for the constructive suggestion to include a summary table. In the revised manuscript, we have added Table1. Cell Interaction Pathways Across Physiological and Pathological States” (Lines 785-786), which not only summarizes aging-related changes but also provides a side-by-side comparison of key pathways across heath, aging, AD, and demyelination. We believe this comprehensive table improves cross-condition comparisons and further enhances the readability of the review.

4. Conclusions Supported by Results

  • Strengths: Conclusions align well with cited evidence, particularly the therapeutic implications of dysregulated glial-neuronal crosstalk. Limitations (e.g., inferential nature of ligand-receptor predictions) are acknowledged.

Response: We appreciate the reviewer’s positive assessment of our conclusions and the acknowledgement of how we addressed study limitations. We have maintained these strengths in the revised manuscript.

5. Figures & Tables Clarity

  • Strengths: Figures (e.g., models of healthy vs. AD/demyelination) are referenced and enhance understanding.
  • Suggestions: A table summarizing key pathways/dysfunctions could consolidate data.

Response: As noted in our response to Comment 3, we have added “Table1. Cell Interaction Pathways Across Physiological and Pathological States” (Lines 785-786), which organizes representative pathways and corresponding citations across health, aging, AD, and demyelination. By consolidating examples such as CX3CL1-CX3CR1, VEGF/Angiopoietin-Tie, JAK/STAT, and TREM2/IL-33/ST2 into a single comparative framework, the table enables rapid cross-condition reference and highlight both shared and distinct mechanisms, thereby enhancing clarity and integration with the figures and narrative.

6. Originality/Novelty

  • Strengths: The review synthesizes emerging spatial transcriptomics data with classic literature, highlighting novel concepts (e.g., APOE4-driven lipid toxicity, SPP1-mediated synapse loss).
  • Highlight: The focus on "communication as a therapeutic target" is forward-thinking.

Response: We are truly grateful for the reviewer’s recognition of the novelty of our synthesis and the forward-looking perspective on “communication as a therapeutic target.” We have preserved these strengths in the revised manuscript.

7. Significance of Content

  • Strengths: The article addresses a timely topic with broad implications for neurodegeneration and repair. The emphasis on glial hubs and spatial signaling is particularly impactful.

Response: We appreciate the reviewer’s positive evaluation of the significance and impact of our work, particularly the emphasis on glial hubs and spatial signaling, which have been retained in the revised manuscript.

8. Quality of Presentation

  • Strengths: Well-structured, with logical flow and professional tone.
  • Minor Issues: A few long sentences could be split for clarity (e.g., Page 1: “ In contrast, single-cell RNA sequencing (scRNA-seq) enables the resolution of gene expression profiles at the level of ndividual cells, uncovering rare or transient cell states and revealing the transcriptional diversity of both neuronal and non-neuronal populations. More recently, spatial transcriptomics techniques, such as Nanostring11, MERFISH12, TF-seqFISH13, 10x Visium and 10x Xenium14, 15, have added a critical spatial dimension to transcriptomic profiling, allowing researchers to map gene expression back to its tissue context without the need for cell dissociation.”).

Response: We thank the reviewer for the constructive suggestion. In line with the recommendation, we have revised the identified sentence to improve clarity by splitting it into shorter, more concise statements (Lines 67-73). Specifically, the original sentence was replaced with:

“In contrast, single-cell RNA sequencing (scRNA-seq) provides gene expression profiles at the level of individual cells. This allows researchers to uncover rare or transient cell states and to reveal the transcriptional diversity across both neuronal and non-neuronal populations. More recently, spatial transcriptomics techniques, such as Nanostring10, MERFISH11, TF-seqFISH12, 10x Visium and 10x Xenium13, 14, have added a critical spatial dimension to transcriptomic profiling, allowing researchers to map gene expression back to its tissue context without the need for cell dissociation.”

In addition, we have revised other long or complex sentences throughout the manuscript to further enhance readability and maintain a clear, professional tone (Lines 134-137, 205-207, 304-306, 362-367, 394-407).

9. Scientific Soundness

  • Strengths: Claims are well-supported by primary studies and computational tools. Balanced discussion of limitations (e.g., lack of protein-level validation).

Response: We thank the reviewer for recognizing the rigor of our evidence and balanced discussion of limitations, which we believe are essential for advancing the field.

10. Interest to Readers

  • Strengths: Highly relevant to neuroscientists, neurologists, and therapeutic developers. The translational focus broadens appeal.

Response: We appreciate the acknowledgement of the manuscript’s broad relevance and translational focus, which were key goals in framing the review.

11. Overall Merit

  • Assessment: Excellent scholarly contribution that advances understanding of CNS communication networks. Meets high publication standards.

Response: We are grateful for the positive assessment of the manuscript’s contribution and quality, and for recognizing its alignment with high publication standards.

12. Quality of English Language

  • Strengths: Clear, concise, and grammatically sound.
  • Minor Edits:
    • Page 1: "transcriptomic" → "transcriptomic analyses" for parallelism.
    • Page 6: "Inflammaging" should be defined at first use.

Response: We thank the reviewer for the positive evaluation of the language quality. We retained the term “transcriptomics” rather than “transcriptomic analyses” in some sentences to preserve the distinction between single-cell and spatial transcriptomics (referring to high-throughput omics approaches) and bulk transcriptomic analyses (which in this context does not denote omics).

As suggested, we have now defined “inflammaging” at its first mention (Lines 257-259) as “These changes shift the tissue microenvironment from a supportive, homeostatic states toward one characterized by chronic low-grade inflammation (“inflammaging”).”

Summary of Recommendations

  1. Add a table summarizing key communication pathways in health/disease.
  2. Define niche terms (e.g., "inflammaging") early.
  3. Consider splitting long paragraphs for readability.

Response: We sincerely thank the reviewer for the thoughtful, constructive feedback and valuable input, which has helped us further enhance the clarity, accessibility, and overall impact of the review.

Recommendation: Minor revision

This is a high-quality, novel review with significant scholarly and translational value. With minor refinements, it would be suitable for a publication in the Int. J. Mol.Sci.

Reviewer 2 Report

Comments and Suggestions for Authors

A Brief Summary

The presented work is a comprehensive review of dynamic intercellular networks in the central nervous system (CNS) with a focus on interaction mechanisms between neurons, astrocytes, microglia, and oligodendrocytes under normal conditions, aging, and neurodegeneration. The main strengths of the work are the integration of modern single-cell and spatial transcriptomics data, systematic description of intercellular communications, and analysis of their disruptions in various pathological conditions. The article makes a significant contribution to understanding the role of intercellular interaction as a therapeutic target.

The bibliography includes predominantly contemporary sources (last 5 years), which meets the requirements for relevance.

General Concept Comments

Comment 1. The structural organization of the "Results" section does not correspond to the review article format. The use of a "Results" section is more suitable for original research. Reorganize the article structure by replacing the "Results" section with thematic subsections with descriptive titles (e.g., "Modes of Intercellular Communication", "Age-related Changes", "Disease-associated Alterations").

Comment 2. The work does not clearly formulate the knowledge gap that this review is intended to fill. In the introduction, more specifically define which aspects of intercellular communication in the CNS remain insufficiently studied and require systematic analysis.

Comment 3. The review covers an extremely broad spectrum of topics, which may reduce its focus. Consider the possibility of narrowing the focus to the most critical aspects of intercellular communication or more clearly structuring the material according to the principle "from general to specific".

All general comments in this case are discussable.

Specific Comments

Lines 23-38: The introduction is too general and does not focus attention on specific aspects of intercellular communication. It would be useful to specify exactly which mechanisms of intercellular communication will be considered and why they are critically important.

Lines 175-234: The section on communication in healthy CNS is well-structured, but the mechanisms are presented fragmentarily. A more systematic description of the hierarchy of intercellular interactions from local to systemic would help the reader navigate the mechanisms of interaction between cells. However, this comment remains at the authors' discretion.

Lines 502-547: The section on therapeutic opportunities is too brief for such an important topic. It seems that expanding the discussion of specific therapeutic strategies with examples of current clinical trials would give the review greater practicality.

Author Response

Reviewer 2:

A Brief Summary

The presented work is a comprehensive review of dynamic intercellular networks in the central nervous system (CNS) with a focus on interaction mechanisms between neurons, astrocytes, microglia, and oligodendrocytes under normal conditions, aging, and neurodegeneration. The main strengths of the work are the integration of modern single-cell and spatial transcriptomics data, systematic description of intercellular communications, and analysis of their disruptions in various pathological conditions. The article makes a significant contribution to understanding the role of intercellular interaction as a therapeutic target.

The bibliography includes predominantly contemporary sources (last 5 years), which meets the requirements for relevance.

Response: We thank the reviewer for the positive evaluation and for recognizing our the strengths of our work, including the integration of modern single-cell and spatial transcriptomics data and the systematic analysis of intercellular mechanisms. We are glad that the review was found to make a meaningful contribution to understanding CNS communication networks and therapeutic targeting.

General Concept Comments

Comment 1. The structural organization of the "Results" section does not correspond to the review article format. The use of a "Results" section is more suitable for original research. Reorganize the article structure by replacing the "Results" section with thematic subsections with descriptive titles (e.g., "Modes of Intercellular Communication", "Age-related Changes", "Disease-associated Alterations").

Response: We appreciate the correction regarding the structural organization of our review. In the revised manuscript, we removed the “Results” section heading, and we replaced it with thematically organized subsections featuring descriptive titles, starting with “2. The Intercellular Communication Landscape in the CNS: Modes and Mechanisms” (Line 88).

Comment 2. The work does not clearly formulate the knowledge gap that this review is intended to fill. In the introduction, more specifically define which aspects of intercellular communication in the CNS remain insufficiently studied and require systematic analysis.

Response: We are grateful for the suggestion regarding the need to more clearly define the knowledge gap addressed by our review. In response, we have revised the introduction to explicitly outline current limitations in the study of CNS intercellular communication. Specifically, we now note that traditional bulk transcriptomic approaches obscure cell-type-specific contributions, spatial organization, and cellular heterogeneity, which has hindered a precise understanding of microenvironmental interactions in the brain. We also added a concluding paragraph in the introduction that clearly states the scope of our review—covering CNS intercellular communications across heathy homeostasis, aging, AD, and demyelination—while emphasizing how single-cell and spatial transcriptomic technologies enable us to identify both shared and unique communication modules and their implications for disease mechanisms and therapeutic strategies (Lines 61-67, 82-86).

Comment 3. The review covers an extremely broad spectrum of topics, which may reduce its focus. Consider the possibility of narrowing the focus to the most critical aspects of intercellular communication or more clearly structuring the material according to the principle "from general to specific".

Response: We thank the reviewer for pointing out the need to sharpen the focus of the manuscript. In accordance with this suggestion, we have restructured the review at both the sectional and sub-sectional levels to more clearly follow a “general to specific” organization. Specifically, sections 2-6 no longer appear under a “Results” heading but are presented as thematically organized units. Each section now begins with broad conceptual principles of intercellular communication and then narrows to specific mechanisms and examples. For instance:

  • Section 2—starts with an overview of the major modes of intercellular communication in the CNS (e.g., ligand-receptor signaling, extracellular vesicles, gap junctions), and then narrows to concrete molecular mediators and representative examples.
  • Section 3—begins with an overview of the technological advances for mapping intercellular communication in the CNS, highlighting the limitations of traditional bulk approaches. It then narrows to specific methods, including single-cell and spatial transcriptomics and integrative multimodal frameworks, which together enable reconstruction of communication networks in both health and disease.
  • Section 4—introduces the broad organizational principles of how intercellular communication sustains homeostasis, and then details specific processes such as synaptic refinement, metabolic coupling, and glia-vascular interactions.
  • Section 5—begins with systemic and general hallmarks of aging that impact communication, before moving to specific examples such as altered astrocyte-microglia crosstalk, changes in neurovascular coupling, and age-dependent shifts in inflammatory signaling.
  • Section 6—begins with a general overview of how intercellular communication is disrupted across CNS pathologies and then proceeds to disease-specific contexts. We first discuss demyelination and remyelination, outlining the general framework of how communication governs repair dynamics, before narrowing to specific mechanisms such as microglia-astrocyte-OPC signaling, lesion compartmentalization, and barriers to remyelination. We then focus on neurodegeneration, with an emphasis on AD, starting with broad principles of dysregulated signaling in neurodegeneration, and then highlighting specific ligan-receptor interactions and spatially organized glial phenotypes as representative examples.

Through this restructuring, the manuscript now emphasizes the most critical aspects of intercellular communication while maintaining a coherent “general to specific” narrative across sections (Lines 88-421)

All general comments in this case are discussable.

Specific Comments

Lines 23-38: The introduction is too general and does not focus attention on specific aspects of intercellular communication. It would be useful to specify exactly which mechanisms of intercellular communication will be considered and why they are critically important.

Response: We thank the reviewer for this constructive comment. In response, we have substantially revised the introduction to clearly specify the scope of our review and the mechanisms of intercellular communication under consideration. We now explicitly state that this review focuses on ligand-receptor-mediated signaling as a central and versatile mode of communication in the CNS, describes its functional significance in both health and disease, and explains why it is critical important for understanding disease progression and therapeutic opportunities. We also highlight the specific knowledge gaps our work addresses—the lack of an integrative, comparative analysis of ligan-receptor signaling networks across multiple CNS states using recent advances in single-cell and spatial transcriptomics (Lines 31-57).

Lines 175-234: The section on communication in healthy CNS is well-structured, but the mechanisms are presented fragmentarily. A more systematic description of the hierarchy of intercellular interactions from local to systemic would help the reader navigate the mechanisms of interaction between cells. However, this comment remains at the authors' discretion.

Response: We thank the reviewer for this insightful suggestion. In response, we have substantially revised the section on intercellular communication in the healthy CNS (Lines 188-254) to provide a more systematic and hierarchical framework, as recommended. Specifically, the revised text now organizes the discussion from local to systematic scales, explicitly covering four key levels of CNS intercellular communication:

  1. Multicellular tissue homeostasis—highlighting coordinated metabolic support, immune surveillance, and barrier integrity maintained by neurons, glia, and vascular cells.
  2. Synaptic refinement—detailing activity-dependent microglia-neuron and astrocyte-neuron signaling that shapes synaptic architecture and function.
  3. Developmental lineage coordination—describing communication between progenitors and differentiated cell types that guides lineage specification and myelination.
  4. Long-range functional integration—outlining vascular-neuronal-glial coupling mechanisms that synchronize activity and homeostasis across brain regions.

In addition, we have integrated recent high-resolution single-cell and spatial transcriptomic findings into each level, thereby linking molecular mechanisms to their spatial and functional contexts. This reconstructing not only narrows the scope to clearly defined mechanisms—particularly ligand-receptor signaling—but also improves the logical flow, enabling readers to navigate from fundamentally local interactions to broader network integration. These changes directly address the reviewer’s comment by defining the precise aspects of intercellular communication under consideration and emphasizing their relevance to CNS physiology.

Lines 502-547: The section on therapeutic opportunities is too brief for such an important topic. It seems that expanding the discussion of specific therapeutic strategies with examples of current clinical trials would give the review greater practicality.

Response: We appreciate the reviewer’s observation that the therapeutic section was previously too brief for such an important topic. In the revised manuscript, we have substantially extended this section (Lines 422-465) to address the concern and to provide a more practical and clinically relevant discussion. Specifically, we incorporated:

  1. Deeper mechanistic framing—we now emphasize that dysregulated intercellular communication is not merely a downstream effect but a primary driver of pathogenesis in neurodegenerative and demyelinating disorders.
  2. Integration of unifying disease principles—the section now synthesizes recent findings into three overarching concepts:
    • Progressive imbalance between homeostatic and pathological signaling.
    • Context-dependent disease phenotypes shaped by local microenvironments.
    • Disruption of spatial organization in intercellular signaling.
  3. Therapeutic strategies anchored in intercellular network restoration—we expanded the discussion to outline strategies that target the restoration of multicellular coordination rather than isolated molecular events, including:
  • Enhancing protective glia-neuron signaling.
  • Reprogramming glial states towards pro-regenerative phenotypes.
  • Stabilizing neurovascular interactions.
  • Spatially targeted drug delivery to lesion-specific niches.
  1. Clinical translation—we strengthened the section by highlighting paradigm shift from single-target interventions to network-level therapeutic approaches, aligning the discussion with emerging opportunities revealed by single-cell and spatial technologies.